# *Enterobius vermicularis* Related Acute Appendicitis: A Case Report and Review of the Literature

**Shabnam Chhetri** [1,*], **Ahmed Hamood Al Mamari** [2], **Mahmood Mausd Al Awfi** [3], **Nasser Humaid Nasser Al Khaldi** [4], **Nibras Mejbel Abed** [3], **Nenad Pandak** [1], **Faryal Khamis** [1], **Zakariya Al Balushi** [1], **Rashid Mohammed Khamis Alalawi** [3], **Sultan Al Lawati** [1], **Muna Ba'Omar** [1], **Nasser Shukaili** [1] and **Seif Al-Abri** [1]

1   Department of Infectious Diseases, The Royal Hospital, Muscat, PC 111, Oman; npandak@gmail.com (N.P.); khami001@gmail.com (F.K.); zak077@gmail.com (Z.A.B.); s.m.h.lawati@gmail.com (S.A.L.); muna.baomar@gmail.com (M.B.); moath201602@gmail.com (N.S.); salabri@gmail.com (S.A.-A.)
2   Department of Microbiology, The Royal Hospital, Muscat, PC 111, Oman; a.h.s.almamari@gmail.com
3   Department of Colorectal Surgery, The Royal Hospital, Muscat, PC 111, Oman; mahmood.alawfi@gmail.com (M.M.A.A.); nibrasmejbel82@gmail.com (N.M.A.); alalawi9999@yahoo.com (R.M.K.A.)
4   Department of Histopathology, The Royal Hospital, Muscat, PC 111, Oman; drnasseralkhaldi@gmail.com
*   Correspondence: drshabnamchhetri24@gmail.com

**Abstract:** While the debate on the association between *Enterobius vermicularis* (*E. vermicularis*) and acute appendicitis has not been settled, a few case reports of this very rare encounter are beginning to come to light. *E. vermicularis* is one of the most common parasitic infections around the world, and acute appendicitis, on the other hand, is also a commonly encountered condition in general surgery. However, the association between these two conditions remains controversial. Here we present a case report of a young woman with appendicitis associated with *E. vermicularis*.

**Keywords:** parasitological diseases; Oman; appendectomy; appendicitis; *Enterobius vermicularis*; enterobiasis; case report; helminths; appendix; pinworms

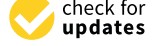



## 1. Introduction

One of the most common emergency abdominal operations worldwide is for suspected acute appendicitis [1]. Appendicitis is defined as an inflammation of the appendix characterized by abdominal pain. It is a malady that occurs when the insides of the appendix are blocked. The diagnosis of acute appendicitis is based on the patient's history, physical examination, laboratory evaluation, and abdominal imaging. The classic symptoms include the aforementioned vague periumbilical pain, as well as anorexia, nausea, intermittent vomiting, migration of pain to the right lower quadrant of the abdomen, and low-grade fever. In approximately 90% of patients presenting with these symptoms, the diagnosis of acute appendicitis has been made. The most common treatment is laparoscopic appendectomy [2]. A wide spectrum of cases have been cited as causes of appendicitis, and among the rarest are neuroendocrine tumors, serrated adenomas, low-grade appendicular mucinous neoplasms, hyperplastic polyps, and intestinal parasites [3].

*Enterobius vermicularis* (*E. vermicularis*) has a worldwide prevalence, though crowded conditions are a risk factor for infection, and they are somewhat more common in developing countries [4,5]. *E. vermicularis*, also known as pinworms, is the most common human parasite, being the most common helminth infection reported in the United States and are an infection transferable only by humans [6]. Among the affected individuals, approximately 40% are asymptomatic in nature. Pinworm infection is actually a self-limiting disease as the life expectancy of adult worms is so short unless an autoinfection occurs. Perianal pruritis is the most common symptom of this infection, which tends to arise the most at nighttime

while a person infected with the parasites is asleep. The itching and discomfort can disrupt sleep, and in some children, in about half of the cases, can cause a loss in bladder control. Scratching the perianal region can lead to ulcerations, and in some cases, the repeated picking of this skin can result in lesions (excoriation), leaving an infected individual vulnerable to bacterial superinfections. Other potential indirect symptoms include perianal folliculitis, anal dermatitis, and ischiorectal abscess. Autoinfection or transmission may occur as a result of scratching as this action itself releases eggs, which can easily stick to the fingers or under the fingernails and be passed on as a result. In some rare cases, pinworms can be indirectly responsible for vulvovaginitis—irritation/inflammation of the area—if pinworms migrate to the vaginal area, and this could also indirectly cause a urinary tract infection as other enterobacteria like *Escherichia coli* can adhere [7]. Pinworm infestation is also considered to be the most common helminthic infection found in the appendiceal lumen—the inside of the appendix—and has been said to cause acute appendicitis.

The role of *E. vermicularis* in relation to the pathogenesis of a few cases of acute appendicitis has been the subject of controversy for many years, as pinworms have been discovered in the appendix as far back as 1634 by Fabrius [7,8], despite the fact that no causality has been reliably demonstrated. The majority of the cases that have been reported so far has mainly been in the pediatric population. We present a case report of an adolescent woman who presented to us with pain in the right iliac fossa. Her abdominal examination revealed tenderness in the right iliac fossa with a positive rebound sign, blood tests showed normal inflammatory markers, and abdominal imaging was suggestive of borderline tip appendicitis (a rare condition where the distal appendix becomes inflamed). Despite adequate conservative management, she continued to remain symptomatic, so she underwent laparoscopic appendectomy. The histopathology of the re-sectioned appendix reported prominent lymphoid hyperplasia, and *E. vermicularis* eggs were found within the lumen of the appendix.

## 2. Case Report

A 21-year-old female with no significant past medical or surgical history presented to the emergency department at the Royal Hospital in Muscat, Oman, complaining of right iliac fossa abdominal pain of a one-week duration. The pain was progressive with associated anorexia. She denied any history of fever or change in her bowel habits but reported having regular clear vaginal discharge before menstruation. Her last menstrual period was a month ago. On examination, her vital signs were within normal limits. Her abdominal examination revealed tenderness in the right iliac fossa with a positive rebound sign. Urine analysis showed that no infection was present and the pregnancy test was negative. Laboratory investigations showed a hemoglobin level of 13.3 g/dL, a white cell count of $4.1 \times 10^9$, and a C-reactive protein level of <4.0 mg/L. The coagulation profile and renal function tests were normal.

The patient's pelvic ultrasonography showed normal ovaries with no free fluid. A computed tomography (CT) scan was performed given the low probability of acute appendicitis (Alvarado score of 4), which showed a retrocecal appendix with a short segment wall thickening of 7 mm and no significant fat stranding suggestive of borderline tip appendicitis. When a reevaluation was performed, abdominal tenderness was persistent despite adequate analgesia and hydration; therefore, a diagnostic laparoscopy and laparoscopic appendectomy under general anesthesia was performed. Intraoperative findings revealed a mild hyperemic appendicular tip, which was adherent to the lateral abdominal wall and cecum with minimal hemorrhagic fluid in the pelvis. Her postoperative period was unremarkable, and she was discharged home after 24 h.

The histopathology report revealed a mild denuded epithelium and prominent lymphoid hyperplasia. In addition, *E. vermicularis* eggs were found within the lumen of the appendix. The patient and her family members received 400 mg of albendazole once weekly for three weeks. On her follow-up appointment, she was feeling well with no complaints (Figures 1–3).

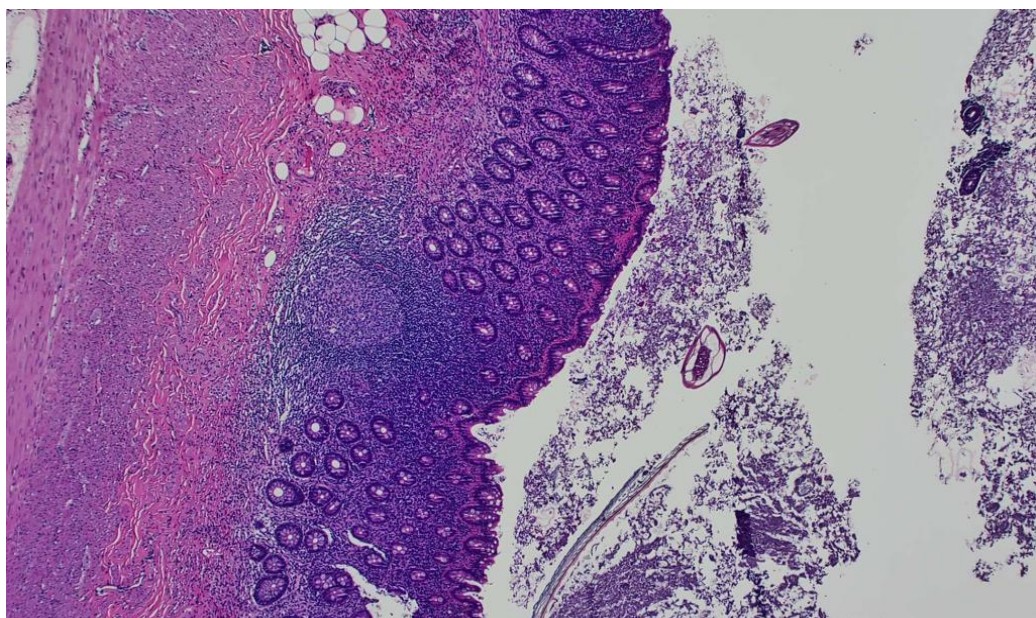

**Figure 1.** Hematoxylin and eosin (H&E) stain showing the appendix mucosa on a 10× view showing extensive lymphoid hyperplasia and intraluminal *Enterobius vermicularis* eggs.

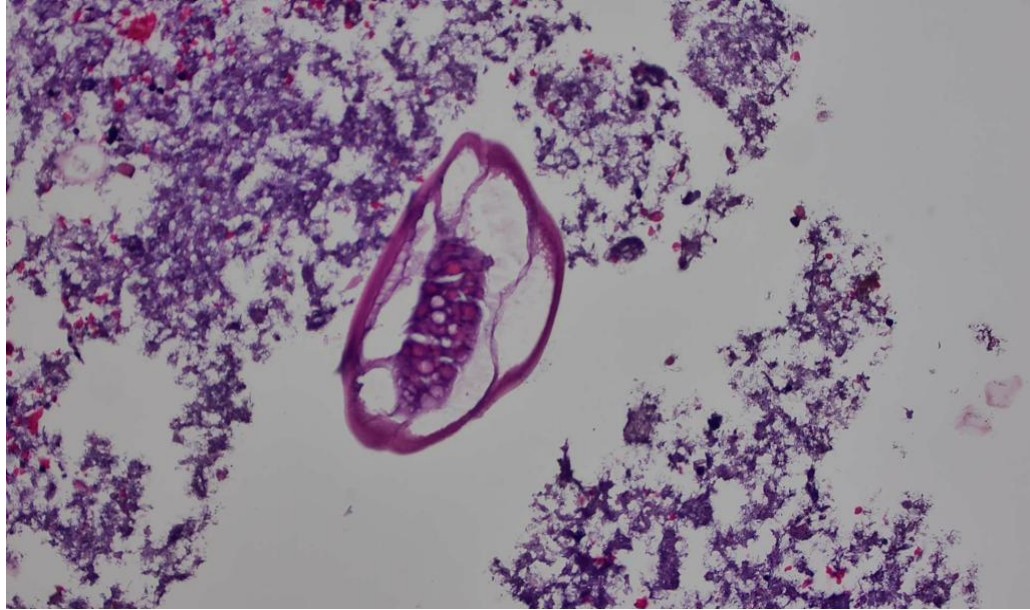

**Figure 2.** H&E stain showing the appendiceal wall on a 40× view showing evidence of *E. vermicularis* infection in the form of *E. vermicularis* eggs.

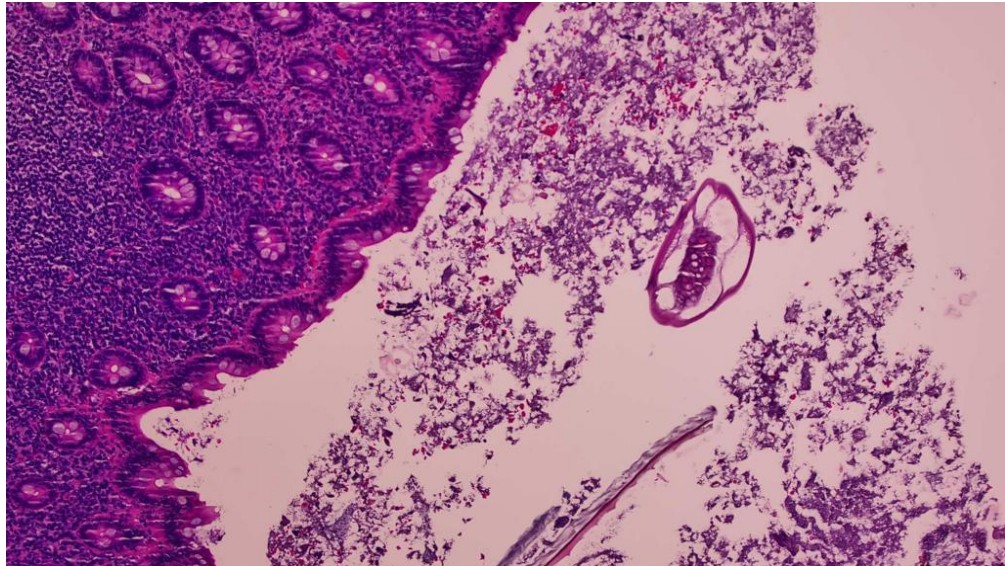

**Figure 3.** H&E stain showing the extensive lymphoid hyperplasia of the appendiceal mucosa on a 20× view, which commonly is associated with *E. vermicularis* infection.

## 3. Discussion

Parasitic infections are a global public health concern. More prominent with higher morbidity and mortality rates in developing countries, *E. vermicularis*, also known as pinworms or *Oxyuris vermicularis*, as they were traditionally called, is one of the most common nematodes helminthiases worldwide, especially among the pediatric group [9]. This infection becomes even more common among institutionalized individuals, and this includes children in daycare settings [6].

Prevalence studies conducted throughout the world in the past decade have shown the rate of *E. vermicularis* infection in children to be around 20% [4,6,10,11]. Rates have been noted as high as 38% in children 5–7 years old in Sri Lanka in 2013 [12], and as low as 0.21% in Taiwan following a 15-year reduction project [13], with 18% noted in a cohort of otherwise healthy Norwegian children [14], and 20% of American children who are said to have had at least one infection during their childhoods, respectively [6].

Even though this least harmful of the gastrointestinal nematode helminths is most common in children, that is not to say that infection does not occur in adults also. Transmission to parents of infected children will occur, and rates among adults who care for children are particularly high [6].

Symptomatic pinworm infections, referred to as enterobiasis, or oxyuriasis, the latter being the older term for the condition, was first described by a Swedish natural scientist named Carl von Linné. The thread-like appearance of these worms is the most distinguishing characteristic feature. The whitish-beige color is also another striking feature, and they are typically round [7].

*E. vermicularis* is acquired typically through the ingestion and occasional inhalation of eggs. The entire life cycle of *E. vermicularis* typically takes 2 to 4 weeks to mature to an adult worm. The time interval between the ingestion of infective eggs and egg-laying, which is also termed as oviposition, is around 2 to 6 weeks. The adult female worm lays the eggs while the host is resting, which is predominantly at night while they sleep. The female worm leaves the anus and affixes the laid eggs to tissue in the perianal area with an adhesive matrix, leading to the most common symptom of enterobiasis—perianal itch. *E. vermicularis* also commonly infects the cecum and rarely, the extra-intestinal organs. The inhabitation of the gastrointestinal tract is considered to be more of an irritation rather than an infection. The most common manifestation is perianal itching, which frequently happens at night, triggered by an inflammatory reaction to the presence of adult worms and eggs on the perianal skin. Itching leads to the disintegration of the worm cuticle,

which leads to a massive release of eggs, which tend to get lodged beneath the fingernails, permitting further autoinfection and person-to-person transmission. Other presentations, such as abdominal discomfort, nausea, and vomiting are generally related to a high worm burden. However, extra-intestinal organ involvement has been reported, including the liver, kidney, peritoneal cavity, uterus, ovaries, mesentery, and lymph nodes [15–18].

The diagnosis of enterobiasis is usually made when a patient gives the history of intermittent perianal pruritis, which is defined as a recurrent but not constant itching and is typical in individuals with pinworm infections. Diagnoses can also be sought through inspection of the clothes and bedsheets, as well as the anal verge, which can demonstrate moving worm-like parasites. In cases of severe infection, the worms may be expelled very visibly within the stool. The identification of the worms macroscopically, or the worms, and/or eggs, microscopically, provides the evidence of infection [7].

The oviposition takes place on the anal folds, with the eggs being incredibly difficult to see with the naked eye due to their small dimensions (50–60 μm × 20–30 μm). They can, however, be swabbed and identified under a microscope without much difficulty. A so-called Scotch tape test, also termed an adhesive cellophane swab, can be performed, even by the infected individual themselves. The test, a sample collection, must be performed first thing in the morning before any washing of the genital area or defecation. Using readily available adhesive tape, with the buttocks spread apart, one can press a small strip several times to the anal and perianal region. The worm eggs are then able to be identified microscopically by affixing the tape, adhesive side down, to a slide [7].

Appendicitis is caused by obstruction of the appendiceal lumen, which leads to the increase in the intraluminal pressure, followed by inflammation caused by bacterial overgrowth, and probably ischemia and rupture without surgical intervention. Fecal stasis, fecaliths, lymphoid hyperplasia, vegetable residues, fruit seeds, and tumors are considered as the most common etiologies of appendiceal lumen obstruction [19,20].

Different intestinal helminths, namely *Ascaris lumbricoides*, *Trichuris trichura*, and *Taenia* species, have been reported as possible causes of acute appendicitis. The most common helminth reported among them is *E. vermicularis* [21]. Studies have shown that *E. vermicularis* was detected histopathologically in 0.3–4% of surgically removed appendixes [22]. In a recent retrospective study conducted in Iran, 20 out of 7628 appendectomy specimens contained *E. vermicularis*, which is a low rate, but one that proves the occurrence [8]. The association between *E. vermicularis* and appendicitis is controversial. Despite years of study into the relationship between *E. vermicularis* and the pathogenesis of appendicitis, how influential this parasite is to induce inflammation is still unclear. Although it may have a role in causing appendiceal discomfort or appendiceal chronic inflammation following obstruction, the majority of cases have not shown acute inflammation. Enterobius infection can cause diseases like acute appendicitis, chronic appendicitis, ruptured appendicitis, gangrenous appendicitis, and perforation, resulting in peritonitis [23]. Nevertheless, the worm or its eggs can obstruct the appendiceal lumen leading to acute appendicitis, which has been observed histologically in some cases. The most common histopathological finding reported in cases of appendicitis with *E. vermicularis* was lymphoid hyperplasia [24].

It has been well established that acute appendicitis is more prevalent in children and adolescents [25], like in our case of a 21-year-old female who presented with acute appendicitis. The histopathology report of the resected appendix showing *E. vermicularis* eggs within the lumen of the appendix was the only pointer towards the cause. Therefore, even though it is rare, *E. vermicularis* should be noted as one of the differential diagnoses for the commonly presented condition of acute appendicitis, as parasitic appendicitis requires a high index of suspicion and can easily be overlooked in low-prevalence regions [26]. Table 1 shows a few recent case reports on the association between *E. vermicularis* and acute appendicitis. These case reports from across the globe reported the incidences of acute appendicitis, the causes of which were attributed to *Enterobius vermicularis* upon the detection of either the parasite at the time of the operation or on the histopathological

examination. The age ranged from 5 to 31 years of age, and the modality of treatment was appendicectomy and the use of anthelminthics.

**Table 1.** The most important reports published in associating *E. vermicularis* to acute appendicitis.

| Authors/References | Year of Publication | No. of Cases | Gender, Mean Age in Years | Treatment | Country |
|---|---|---|---|---|---|
| Flores [27] | 2022 | 1 | Male, 10 | Laparoscopic appendicectomy/anthelminthic | Mexico |
| Alshihmani [28] | 2022 | 1 | Male, 17 | Open appendicectomy/antihelminthic | Iraq |
| Hooshyar [22] | 2021 | 1 | Female, 8 | Open appendicectomy | Iran |
| Antilahy [29] | 2021 | 1 | Female, 5 | Laparoscopic appendicectomy/anthelminthic | France |
| Lala [30] | 2014 | 109 | NA, 11.4 | Laparoscopic appendicectomy/anthelminthic | New Zealand |
| Akkapulu [24] | 2015 | 9 | 7 females, 2 males, 31 | Appendectomy | Turkey |
| Fleming [31] | 2013 | 13 | NA, 11 | Appendectomy | Ireland |
| Hamdona [32] | 2013 | 30 | 17 male, 13 female, NA | Appendectomy | Palestine |
| Upadhyaya [33] | 2015 | 7 | 4 females, 3 males, 25 | Appendectomy | Nepal |

Laparoscopic appendectomy remains the treatment of choice for acute appendicitis. However, evidence has been mounting towards the use of broad-spectrum antibiotics, such as piperacillin–tazobactam monotherapy or combination therapy with either cephalosporins or fluoroquinolones with metronidazole, which has successfully treated uncomplicated acute appendicitis in approximately 70% of patients. The imaging findings have been sorted to categorize these patients, with CT showing an appendiceal dilatation (appendiceal diameter $\geq$ 7 mm), or the presence of appendicoliths, which has been defined as the conglomeration of feces in the appendiceal lumen and was identified patients for whom an antibiotics-first management strategy was more likely to result in failure. Therefore, for these group of patients who are fit for surgery, defined as having a relatively low risk of adverse outcomes or postoperative mortality and morbidity, appendicectomy was recommended. But in patients without the above-mentioned high-risk CT findings, either appendectomy or antibiotics were considered as the first-line therapy. Even in those patients with high-risk CT findings, who were rendered as being unfit for surgery, the antibiotics-first approach was recommended, and surgery was considered if the antibiotic treatment were to fail [3]. Similarly, in cases involving enterobiasis infection, treatment with anthelminthic medication for the patient after appendectomy has been highly essential. Enterobiasis is a very common and contagious infection, and which has a high cure rate, but with very common recurrences. This infection can spread easily among family members due to the long viability of its eggs on clothes, which can linger on material for around two to three weeks. With the diagnostic challenges and asymptomatic inhabitation of *E. vermicularis*, the need to treat close household contacts to prevent the transmission and possible complications as stated above, is however, an unexplored field.

Hence, to conclude, though real, genuine appendicitis can still potentially be caused by *E. vermicularis* infection in the right circumstances where the patient has a high worm burden that can cause appendiceal lumen obstruction and eventually acute appendicitis, the mere presence of *E. vermicularis* alone in the appendix may otherwise only cause a local irritation, which leads to lymphoid hyperplasia but not diagnosable appendicitis. Enterobiasis can mimic the clinical picture of acute appendicitis without the inflammation of the appendix. As observed in this case, most patients who presented with a clinical picture of acute appendicitis where *E. vermicularis* was found in their appendix had normal

inflammatory markers, and their appendix showed lymphoid hyperplasia. This could imply that the presence of *E. vermicularis* might actually only mimic acute appendicitis. There is perhaps a need to assess for the presence of *E. vermicularis* prior to appendectomy, especially if the patient has normal inflammatory markers and is clinically stable. This could lead to the avoidance of unnecessary surgeries and their possible associated complications by simply treating their *E. vermicularis* infection. However, this is just a possibility, and this theory requires further discussions in the future to come to a unified protocol on how to treat cases of suspected appendicitis.

**Author Contributions:** Conceptualization, original draft preparation: S.C., A.H.A.M. and M.M.A.A.; Images and descriptions: N.H.N.A.K.; Surgery and follow-up: M.M.A.A. and N.M.A.; review and editing: S.A.-A., N.P., F.K., Z.A.B., R.M.K.A., S.A.L., M.B. and N.S. All authors have read and agreed to the published version of the manuscript.

**Funding:** This research received no external funding; did not receive any specific grant from public, commercial, or not-for-profit funding agencies.

**Institutional Review Board Statement:** The study was conducted in accordance with the Declaration of Helsinki, and approved by the Institutional Review Board (or Ethics Committee) of The Royal Hospital (protocol code CR#2023/26).

**Informed Consent Statement:** Informed consent was obtained from the patient involved in the study.

**Data Availability Statement:** Data is unavailable.

**Acknowledgments:** The authors wish to thank Lesley Carson for her editorial assistance in finalizing the manuscript for publication.

**Conflicts of Interest:** The authors declare no conflict of interest.

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
