# Peer review of "Enterobius vermicularis Related Acute Appendicitis: A Case Report and Review of the Literature"

_2036-7449, doi:10.3390/idr15040042_

Round 1

Reviewer 1 Report

Appendicitis is a known complication of pinworm (Enterobius vermicularis) infection. The article under review adds to the evidence in the literature of the presence of this parasite during appendicitis. The article contains interesting information and can be published following minor revision.

Specific comments:

Page 2, line 68-93: The authors describe the young woman’s symptoms, but there is no clear information as to whether she had recently suffered any parasitic infection, including whether she or anyone in her family had been infected with E. vermicularis. Did the authors ask about this, or do they have information from the doctors who were treating the patient?

All photographs must be correctly labelled. The authors did not indicate which photo is a, b and c.  

Discussion

The discussion is interesting and it is good that it contains information about the biology of the pinworm, but this part could be a bit shorter.

The part of the discussion about the relationship between pinworm infection and appendicitis it too superficial and is simply a discussion of the results, without a review of the literature. The data presented in Table 1 are interesting, but the information about these studies should be discussed in the text, and there should be a short description of the results or the conclusions derived from the works cited in the table.

References

The references are well chosen, but seem sparse for this topic. For example, why didn’t the authors cite Zuhair D. Hammood ‘Enterobius vermicularis causing acute appendicitis, a case report with literature review’ International Journal of Surgery Case Reports, 63, 2019, pages 153-156, a similar work describing the case of a 23-year-old woman (https://doi.org/10.1016/j.ijscr.2019.09.025), or several others describing acute appendicitis with Enterobius vermicularis? How did the authors choose which cases to present in Table 1?

Author Response

Reviewer’s comment Point 1:The authors describe the young woman’s symptoms, but there is no clear information as to whether she had recently suffered any parasitic infection, including whether she or anyone in her family had been infected with E. vermicularis. Did the authors ask about this, or do they have information from the doctors who were treating the patient?

Response: Upon admission these questions were not addressed but upon the release of the histopathology report, on her review at the clinic the patient was asked these questions, there was no definite history of Pinworm infection.

Point 2: All photographs must be correctly labelled. The authors did not indicate which photo is a, b and c.

Response : labelling of the photographs have been done.

Point 3: The discussion is interesting and it is good that it contains information about the biology of the pinworm, but this part could be a bit shorter.

Response: The information about the biology of the pinworm has been shortened.

Point 4: The part of the discussion about the relationship between pinworm infection and appendicitis it too superficial and is simply a discussion of the results, without a review of the literature.The data presented in Table 1 are interesting, but the information about these studies should be discussed in the text, and there should be a short description of the results or the conclusions derived from the works cited in the table.

Response: Further data on the controversial relationship between pinworm and acute appendicitis has been added. A short description of the table has been added.

Point 5: The references are well chosen, but seem sparse for this topic. For example, why didn’t the authors cite Zuhair D. Hammood ‘Enterobius vermicularis causing acute appendicitis, a case report with literature review’2264 1407.

Response: Thank you for your comment on our case report. We appreciate your feedback and would like to address your concerns.

Firstly, we apologize if there was any confusion regarding the purpose of our article. As mentioned, our intention was to present a case report rather than provide a comprehensive review on the topic. Case reports typically focus on describing a specific case and its unique aspects, rather than extensively discussing existing literature.Regarding Table 1, we included it to provide a concise summary of the main published case reports related to our topic. We understand your suggestion of discussing the information from the table in the text itself and providing a brief description of the results or conclusions derived from the works cited. This could indeed enhance the discussion section by providing additional context and highlighting key findings. We appreciate your suggestion and will consider incorporating it into future revisions of our article.

Lastly, regarding the discussion on the relationship between pinworm infection and appendicitis, we apologize if it came across as too superficial. We acknowledge that a more comprehensive review of the literature could have been beneficial. While our focus was primarily on presenting the case and its implications, we understand the importance of discussing the broader context and available evidence. In future revisions, we will strive to provide a more in-depth discussion by reviewing relevant literature on the topic.Once again, we thank you for your valuable feedback. Your suggestions will help us improve the clarity and quality of our case report.

6: How did the authors choose which cases to present in Table 1?
Response: we choose amongst case reports that were recent and those we could have access too.

Reviewer 2 Report

The manuscript presents, in my opinion, an important question about the presence of E. vermicluaris in the appendix and the need or not to proceed with an appendectomy.

  I suggest the following changes:

1. Line 33: What is poyp? Please correct.

2. Line 59 to 67: Starting at Here, .... I suggest removing this text part as it is anticipating the case report.

  Figure 1: The identification letter must be added to each photo, in addition to the optical scale used.

Line 110: Italicize E. vermicularis

Line 152: After the text I suggest adding some reference.

Line 168: After the sentence about the parasites, I suggest putting a reference

Bibliography : Spell species names in italics

Author Response

Point 1: Line 33: What is poyp? Please correct
Response: It has been crosschecked as polyp.

Point 2:Line 59 to 67: Starting at Here, .... I suggest removing this text part as it is anticipating the case report.

Response: the text has been removed.

Point 3: Figure 1: The identification letter must be added to each photo, in addition to the optical scale used.

Response: Each photo has been labelled and the optical scale used.

Point 4: Line 110: Italicize E. Vermicularis.
Response: It has been italicized.

Point 5: Line 152: After the text I suggest adding some reference.
Response: reference has been added.

Point 6: Line 168: After the sentence about the parasites, I suggest putting a reference.

Response : The reference is included in 16-19.

Point 7: Bibliography : Spell species names in italics.

Response: the names have been italicized.

Reviewer 3 Report

The relevance of this MS is to draw the attention for enterobiasis and related pathological consequences that are rarely reported, as the present case of acute appendicitis. Overall, the clinical case well described and the MS well written and clear to the readers. I suggest some minor issues that should be done.

- Title: Enterobius vermicularis – specie in lower-case letter

- Abstract, Line 14: “…parasitic infections” - infestation refers to ectoparasites. Please, correct all over the text (ie. Lines 52, 103, 150 …)

- Line 51: “…to the vaginal area…”

- Fig. 1b – In this photo, there is no visible E. vermicularis adult worm; high power view is of the egg.

- Line 110 – name of the specie in italic

- Line 153 – “Since the oviposition takes…” is not the justification for the invisibility of the eggs; the reason for the difficulty to see E. vermicularis eggs is due to their small dimensions (50-60 µm x 20-30 µm, CDC-DPDx) only visible by microscopy.  The sentence should be rephrased.

Author Response

Reviewer's comment Point 1: Title: Enterobius vermicularis – specie in lower-case letter.

Response: It has been corrected.

Point 2: Abstract, Line 14: “...parasitic infections” - infestation refers to ectoparasites. Please, correct all over the text (ie. Lines 52, 103, 150 ...)

Response: it has been corrected.

Point 3: Line 51: “...to the vaginal area...”

Response : correction has been made.

Point 4: Fig. 1b – In this photo, there is no visible E. vermicularis adult worm; high power view is of the egg.

Response: it has been corrected.

Point 5: Line 110 – name of the specie in italic.

Response: it has been italicized.

Line 153 – “Since the oviposition takes...” is not the justification for the invisibility of the eggs; the reason for the difficulty to see E.vermicularis eggs is due to their small dimensions.The sentence should be rephrased.

Response: it has been rephrased.